# Development of Genomic Resources in Mexican *Bursera* (Section: *Bullockia:* Burseraceae): Genome Assembly, Annotation, and Marker Discovery for Three Copal Species

**DOI:** 10.3390/genes13101741

**Published:** 2022-09-27

**Authors:** Yessica Rico, Gustavo P. Lorenzana, Carlos A. Benítez-Pineda, Bode A. Olukolu

**Affiliations:** 1Instituto de Ecología, A.C., Red de Diversidad Biológica del Occidente Mexicano, Pátzcuaro 61600, Mexico; 2Independent Consultant, Pátzcuaro 61608, Mexico; 3Department of Entomology and Plant Pathology, University of Tennessee, Knoxville, TN 37996, USA

**Keywords:** whole-genome sequencing, genomic resources, OmeSeq-qRRS (quantitative reduced-representation sequencing), tropical dry forest, Mexico

## Abstract

*Bursera* comprises ~100 tropical shrub and tree species, with the center of the species diversification in Mexico. The genomic resources developed for the genus are scarce, and this has limited the study of the gene flow, local adaptation, and hybridization dynamics. In this study, based on ~155 million Illumina paired-end reads per species, we performed a de novo genome assembly and annotation of three *Bursera* species of the *Bullockia* section: *Bursera bipinnata, Bursera cuneata,* and *Bursera palmeri*. The total lengths of the genome assemblies were 253, 237, and 229 Mb for *B. cuneata*, *B. palmeri*, and *B. bipinnata*, respectively. The assembly of *B. palmeri* retrieved the most complete and single-copy BUSCOs (87.3%) relative to *B. cuneata* (86.5%) and *B. bipinnata* (76.6%). The ab initio gene prediction recognized between 21,000 and 32,000 protein-coding genes. Other genomic features, such as simple sequence repeats (SSRs), were also detected. Using the de novo genome assemblies as a reference, we identified single-nucleotide polymorphisms (SNPs) for a set of 43 *Bursera* individuals. Moreover, we mapped the filtered reads of each *Bursera* species against the chloroplast genomes of five Burseraceae species, obtaining consensus sequences ranging from 156 to 160 kb in length. Our work contributes to the generation of genomic resources for an important but understudied genus of tropical-dry-forest species.

## 1. Introduction

The development of genomic resources, such as the assembly of reference genomes and their annotations, is still rare for nonmodel organisms of tropical forest ecosystems, which are hotspots of global biodiversity. The availability of species genomes facilitates a wide range of ecological and evolutionary studies, including insights into candidate genes under selection and local adaptation, assessments of the genome-wide diversity and divergence, the characterization of the gene composition and function, marker discovery, among others [1,2,3]. Reference genomes provide a basis for subsequent population genomic studies, which can inform the conservation status and trends of vulnerable species [4,5].

*Bursera* comprises a taxonomic genus of approximately 100 deciduous and resinous tree and shrub species, with a distribution range that spans from the southwestern United States to Peru, the Bahamas, the Galapagos, and the Greater Antilles. The genus has its center of species diversity and endemicity in the Balsas Basin of western Mexico, and it is a distinctive element of the tropical dry forests (TDFs) [6]. *Bursera* species are good candidates for the restoration of disturbed TDF sites because of their capacity for asexual propagation through rooted cuttings [7]. Many *Bursera* species have also had economic and cultural importance since pre-Columbian times. Its aromatic resins are used for religious and medicinal purposes, while the wood is utilized for the elaboration of handcrafts [8], and thus they are subject to selective logging in some rural regions. Due to their overexploitation and the increasing loss of the TDF by anthropogenic land-use changes [9,10], 67% of *Bursera* species are under some threat category within the IUCN Red List [11].

Genetic studies on *Bursera* are scarce, and most of them are focused on its phylogenetic relationships [12,13,14,15], with only some examples on the phylogeography [16,17], population genetics [18], interspecific hybridization [19,20], and marker discovery for taxonomic applications [21]. The genome assembly and annotation in Mexican *Bursera* would facilitate an understanding of the patterns of the local adaptation, genome-wide diversity and differentiation, and hybridization dynamics. We report the first de novo genome assemblies for three copal species of the *Bullockia* section, which are widespread in Mexico: *B. bipinnata, B. cuneata,* and *B. palmeri* [22]. The three species are diploids [23]. *Bursera bipinnata* has the widest distribution across the TDF along the Pacific coast of Mexico, with an altitudinal interval from 1650 to 2200 masl. *Bursera palmeri* is distributed at the southwest of the Mexican Plateau, with an altitudinal interval from 1600 to 2300 masl, while *B. cuneata* occurs mostly in the Bajío region, with an altitudinal interval from 1850 to 2500 masl. The three species occur in sympatry in the remnant TDFs of Michoacán and Guanajuato, where they exchange gene flows and produce fertile hybrids [20]. *Bursera bipinnata* is the most common species that is commercially exploited for the extraction of aromatic resins. *Bursera palmeri* is used locally as a fuel source, while the wood of *B. cuneata* is utilized for the elaboration of religious figures and traditional masks in Michoacán.

The main aim of this work was to generate draft reference genomes through de novo genome assembly and the annotation of *B. bipinnata, B. cuneata,* and *B. palmeri.* We also mapped the obtained filtered reads against the chloroplast genomes of five Burseraceae species: *Boswellia sacra* [24], *Canarium album* [25], *Commiphora wightii* [26], *C. gileadensis*, and *C. foliacea* [24]. Additionally, for highlighting the utility of these genomes for subsequent population genetic and genomic studies, we identified nuclear genetic markers, such as simple sequence repeats (SSRs), in each of the assemblies, and single-nucleotide polymorphisms (SNPs) in a group of 43 *Bursera* individuals of the three species, using the assemblies as a reference. Our work thus contributes to the generation of genomic resources for an important genus of tropical-dry-forest trees.

## 2. Materials and Methods

### 2.1. Sampling and Sequencing

To accomplish the genome sequencing, we collected leaf tissue from single individuals in the localities where each of the species occurs in allopatry (*B. bipinnata*: Jalisco; *B. palmeri*: Querétaro; *B. cuneata*: Ciudad de México) to avoid sampling potentially introgressed individuals resulting from interspecific hybridization [20]. Additionally, we collected leaf-tissue samples from 43 *Bursera* trees (*B. cuneata*: *n* = 23; *B. palmeri*: *n* = 4; *B. bipinnata*: *n* = 16), which were used to identify SNPs using the OmeSeq-qRRS (quantitative reduced-representation sequencing) method (see below). Most samples were from the state of Michoacán (Appendix A). Leaves were preserved in sealable plastic bags containing silica gel until DNA extractions were performed.

Genomic DNA from 20 mg of dried tissue were extracted using the CTAB extraction protocol with prewash steps to eliminate the excess polyphenols [27]. DNA integrity was evaluated by observing a unique DNA band through 1% agarose gel (i.e., not multiple bands). Yield was estimated using a fluorimeter method according to the manufacturer’s instructions (Qubit; Thermo Fisher Scientific, Waltham, MA, USA). For the whole-genome sequencing, 100 ng of high-molecular-weight DNA was used for the Illumina Truseq Nano DNA Library Prep Kit (Illumina, San Diego, CA, USA), according to the company’s guidelines. The resulting libraries were subsequently sequenced in two lanes of the Illumina NovaSeq S4 flow-cell system to obtain ~140–170 million reads (150 bp paired-end reads) per each species.

We used 25 ng of high-molecular-weight DNA, which was extracted from the 43 *Bursera* individuals. DNA was sequentially double digested with the restriction enzymes NsiI-HF and NlaIII. Barcoded adapters were incorporated into genomic fragments following the OmeSeq-qRRS method [28]. Resulting libraries were then diluted to 10 nmol/L for sequencing on a single lane of the NovaSeq S4 flow-cell system (150 bp paired-end reads). The whole-genome and OmeSeq library preparations and sequencing were performed by the NC State Genomic Sciences Laboratory (Raleigh, NC, USA).

### 2.2. Draft Genome Assembly and Annotation

Raw reads from whole-genome sequencing were preprocessed to eliminate adapters and low-quality bases from paired-end short reads using FastQC [29] and Fastp [29,30]. We performed preliminary runs with three de novo genome-assembly methods using their respective default parameters, with *k-mer* sizes ranging from 75 to 90: ABySS v.2.3.4 [31], SGA [32], and Platanus v.1.2.4 [33]. After the preliminary runs, we selected ABySS as the best method based on the computer run time, contig size, and BUSCO completeness. Assembly optimization using ABySS was then performed by modifying the *k-mer* size (*k*) and *k-mer* minimum coverage multiplicity cutoff (*kc*) to obtain the largest N50 and BUSCO (Benchmarking Universal Single-Copy Orthologs) completeness metrics, evaluated against the Eudicotyledoneae-lineage database (eudicots_odb10), downloaded from BUSCO v.3.0.2 [34]. We then proceeded to close the gaps between the contigs for each of the selected assemblies (Fasta format) using the ABySS-sealer module and BESST [35]. Genome features, such as genome size and single-copy regions, were estimated using the *k-mer* method [36,37], which relies on counting the total nonerroneous *k-mers* in the filtered raw reads. For this end, we used Jellyfish v.2.3.0 [38], and we averaged the results for the odd-number words ranging from 17 to 31. Consequently, we performed the contiguity test in QUAST v.5.0.2 [39] and BUSCO completeness analyses to obtain the definitive metrics for each assembly.

For the annotation process, we ran the pipeline in the GenSAS (Genome Sequence Annotation Server) online platform [40]. Due to constraints imposed for uploading data to the server, we performed the annotation on subsets of the assemblies, which encompassed those contigs larger than 2.5 kbps. The ab initio gene prediction was automated through the *Structural* module in GenSAS, using AUGUSTUS v.3.4.0 [41], with the parameter settings for: *Arabidopsis thaliana* species annotation, to report genes on both strands, and to predict complete genes, using the GeneMark-ES v.4.48 self-training algorithm for novel gene identification [42], and specifying a minimum contig length of 50 kbp, and a maximum gap of 5 kbp.

Using BWA-MEM [43], we mapped the preprocessed filtered reads against the genome assembly of *Bowsellia sacra* (NCBI: GCA_013180625.1), which is an African tree of the Burseraceae family, and which is the closest species to *Bursera* with an available whole-genome sequence [44]. Additionally, we aligned the preprocessed filtered reads to the chloroplasts of five species belonging to Burseraceae: *B. sacra* (NCBI: NC_029420.1); *C. album* (NCBI: NC_048982.1); *C. foliacea* (NCBI: NC_041103.1); *C. gileadensis* (NCBI: NC_041104.1); *C. wightii* (NCBI: NC_036978.1).

### 2.3. Marker Discovery

The *SSR Finder* module in GenSAS [41] was employed to identify dinucleotide to hexanucleotide SSRs for each species in their respective assemblies. We specified three repetitions for di, tri, and tetranucleotides: four in pentanucleotides motifs, and five in hexanucleotide motifs.

To identify the SNPs, the OmeSeq raw reads for the 43 *Bursera* samples were demultiplexed and quality filtered for removing erroneous base calls and contaminating adapter sequences using the automated pipeline ngsComposer [45]: https://github.com/bodeolukolu/ngsComposer; accessed on 16 May 2022). Subsequently, the assembled genomes of the three *Bursera* species were used as a reference for the variant calling and filtering with the automated pipeline GBSapp (https://github.com/bodeolukolu/GBSapp; accessed on 16 May 2022). SNPs with minor allele frequencies (MAF) lower than 0.02 were removed, and the SNPs had to be called in at least 80% of the individuals.

To observe the genetic relationships between the three species, we performed a discriminant analysis of principal components (DAPC) in the R package *Adegenet* [46]. The DAPC is a multivariate analysis that is free of HW and LD assumptions, which maximizes the genetic variation among groups, which, in our case, are the species. The optimal number of PCs to retain was optimized with the *xvalDapc* function [46].

## 3. Results

### 3.1. Genome Assembly and Annotation

A total of 34.5 Gb of raw 151 bp Illumina reads were sequenced for the tree copal species. The resulting draft genome of *B. cuneata* was 253 Mb in size, with a GC content of 34.97%, and an N50 length of 0.014 Mb. The assembled genome size of *B. bipinnata* was 229 Mb, with a GC content of 34.14%, and an N50 length of 0.006 Mb. The assembled genome size of *B. palmeri* was 237 Mb, with a GC content of 34.11%, and an N50 length of 0.011 Mb. The assembly statistics are shown in Table 1. The genome assembly of *B. palmeri* recovered the most complete and single-copy BUSCOs (87.3%) relative to *B. cuneata* (86.5%) and *B. bipinnata* (76.6%). The completeness assessments of the three assembled genomes are shown in Figure 1.

The estimated genome sizes based on the *k-mer* method applied to the filtered reads were 565 ± 189, 591 ± 172, and 840 ± 80 Mb for *B. cuneata*, *B. palmeri*, and *B. bipinnata*, respectively. Similarly, the estimated single-copy regions were 301 ± 117, 290 ± 94, and 351 ± 71 Mb, respectively, implying that the final assemblies recovered between 65% and 84% of the nonrepetitive regions along the genomes. This allowed for the identification of 31,357 protein-coding genes in *B. cuneata*, 29,558 in *B. palmeri*, and 21,043 in *B. bipinnata* through the ab initio prediction approach.

The alignment and mapping of the filtered reads against the nuclear genome of *B. sacra* resulted in approximately 53–63 million mapped reads, which represented ~40% of the total filtered reads in each piece of the *Bursera* sequencing data (Table 2). Similarly, the identification of the chloroplast genomes resulted in between approximately 3.9 and 7.5 million mapped reads against the five reference genomes, which represents from ~2.8 (*B. bipinnata*) to 4.4% (*B. cuneata*) of the total filtered reads. *C. wightii* was the reference plastid genome with the most matching reads against the three *Bursera* species (Table 3). The chloroplast genomes were 156–160 kb in length, with a GC content of ~27%.

### 3.2. SSR and SNP Discovery

A total of 107,270, 100,614, and 76,766 dinucleotide to hexanucleotide SSRs were identified for *B. cuneata*, *B. palmeri*, and *B. bipinnata*, respectively (Appendix A). For the three species, dinucleotides were the most common motifs, accounting for ~36% of all the identified SSRs, followed by the tetranucleotide (~29%) and trinucleotide motifs (~19%). Smaller numbers of pentanucleotide (~9%) and hexanucleotide (~4%) motifs were also observed among the three species (Table 4). Within the dinucleotide motifs, the AT/TA were the most frequent, accounting for ~65% of the motifs identified in this category, while, for the tetranucleotides, the AAAT/TTTA motifs were the most common (~20%); these trends were shared among the three species.

The alignment and mapping of the OmeSeq filtered reads against the three draft genomes in a set of 43 *Bursera* individuals resulted in a high proportion of reads (i.e., from 95 to 99%) that mapped to the reference genomes. The variant calling and filtering resulted in 5543 biallelic SNPs based on a minor-allele-frequency (MAF) threshold of 0.02, and no more than 20% missing SNPs across individuals. The results from the DAPC showed the distinction among the three species (Figure 2), although there was a large overlap between some individuals of the three species: *B. cuneata* and *B. bipinnata*, being the two species with greater genetic similarity.

## 4. Discussion

The availability of sequenced genomes is lacking for the genus *Bursera*, despite its ecological and cultural importance in neotropical-dry-forest ecosystems. So far, the only available genome within the family Burseraceae is for *Boswellia sacra*, which occurs in Asia and Africa. Here, we generated the first de novo genome assemblies for three *Bursera* species of the *Bullockia* section: *B. bipinnata*, *B. cuneata*, and *B. palmeri*, which are native to Mexico.

Even though the draft genomes were assembled by relying on short paired-end Illumina reads, the quality obtained was acceptable (BUSCOs ranging from 76 to 87%), and we could recover between 21,043 and 31,357 genes based on the ab initio gene prediction. The lower contiguity and BUSCO scores obtained for *B. bipinnata* relative to the other two *Bursera* species were likely associated with the quality of the DNA and its larger estimated genome size (it is noteworthy that *Bursera* species are diploids, and they have small genomes compared with other forest tree species [23]). The estimated genome sizes for the three *Bursera* species based on the *k-mer* count ranged from 565 to 840 Mb, which are larger than the genome assembly size reported for *B. sacra* (432 Mb). The read mapping of the three *Bursera* species against the nuclear genome of *B. sacra* was relatively low (<50%), likely reflecting their large phylogenetic distance within the Burseraceae family [13]. This highlights the importance of developing genus- and species-specific genomic resources for *Bursera*, as the available genomic data in other genera within the Burseraceae family are not representative of the genome-wide variation in *Bursera*. It is especially important to develop draft genomes for species within each of the two *Bursera* sections, as they have distinct evolutionary histories [47].

Moreover, we identified thousands of nuclear SSR markers in each species, the most common being the dinucleotide repeats. Specifically, the AT/TA motifs were by far the most common, which agreed with previous studies on SSR discovery in plant species [48,49]. SSRs are cost-effective and highly variable codominant markers that are very popular in conservation genetic studies [50]. The lack of available polymorphic markers, such as SSRs, in species of *Bursera* has precluded the implementation of intraspecific genetic studies for assessments of the genetic diversity, gene flow, and structure in vulnerable species. A subsequent step of the SSRs identified here would be their PCR validation and assessments of the levels of polymorphisms. Such polymorphic markers could then be potentially cross amplified in closely related *Bursera* species, thus contributing to increase our knowledge on the population genetic diversity and structure in several species.

Additionally, to exemplify the utility of our nuclear genome assemblies as reference genomes, we performed the identification of biallelic SNPs in a set of 43 *Bursera* individuals, independently sequenced through the OmeSeq-qRRS approach. The genome-wide markers that were identified constitute a valuable tool to evaluate the neutral and adaptive genetic variation for intra- and interspecific-level comparisons in genomic studies. In our case, the DAPC not only showed the species clustering, but also the overlapping among the individuals of the three species. These three *Bursera* species are known to hybridize in co-occurring locations [20], which may explain the observed intermixing resulting from the interspecific gene flow. The development of genomic resources for three Mexican *Bursera* species is a first step toward increasing the scientific investigations on population genomics and comparative phylogenetic studies.

## Figures and Tables

**Figure 1 genes-13-01741-f001:**
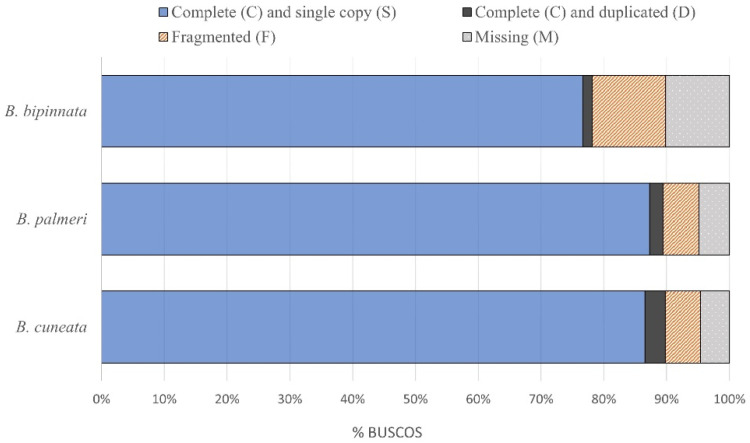
Results from three *Bursera* genome contiguity assessments using BUSCO.

**Figure 2 genes-13-01741-f002:**
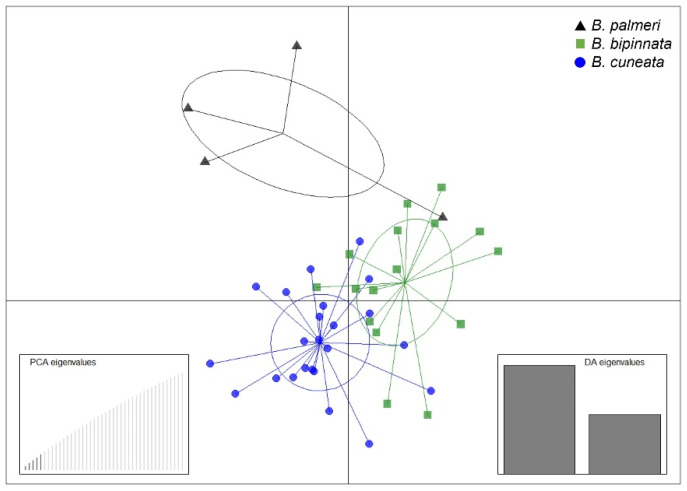
DAPC plot based on 5543 biallelic SNPs for 43 *Bursera* individuals. The left inset shows the number of retained PCs, and the right inset denotes the first two discriminant functions used.

**Table 1 genes-13-01741-t001:** Assembly metrics for three *Bursera* de novo nuclear genomes.

Species	Genome Assembly Size (Mb)	Contigs (n)	N50 (bp)	L50 (n)	Largest Contig (bp)	GC Content (%)
*B. cuneata*	252.9	70,005	13,532	4324	203,078	34.97
*B. bipinnata*	229.2	93,723	5699	9652	104,624	34.14
*B. palmeri*	237.4	67,240	11,112	4918	230,450	34.11

**Table 2 genes-13-01741-t002:** Alignment and mapping of the preprocessed filtered reads of the three *Bursera* species against the nuclear genome of *Bowsellia sacra*.

Species	Total Primary Reads (Million)	Mapped Reads (Million)	Properly Paired Reads (Million)
*B. cuneata*	169.8	62.9 (37.0%)	43.5 (25.6%)
*B. bipinnata*	141.9	53.6 (37.8%)	35.6 (25.1%)
*B. palmeri*	145.2	62.9 (43.4%)	48.6 (33.5%)

**Table 3 genes-13-01741-t003:** Alignment and mapping of the preprocessed filtered reads of the three *Bursera* species against the chloroplast genomes of five Burseraceae species. The percentages of the total mapped filtered reads are in parentheses.

Species	Reads (M)	*Boswellia sacra*	*Canarium album*	*Commiphora foliacea*	*Commiphora gileadensis*	*Commiphora wightii*
*B. cuneata*	169.8	7,481,910 (4.40%)	7,253,921 (4.27%)	7,304,511 (4.30%)	7,364,709 (4.34%)	7,514,759 (4.42%)
*B. bipinnata*	141.9	4,080,000 (2.87%)	3,908,183 (2.75%)	3,948,175 (2.78%)	3,970,894 (2.80%)	4,122,751 (2.90%)
*B. palmeri*	145.2	5,187,716 (3.57%)	4,953,793 (3.41%)	5,043,110 (3.47%)	5,042,119 (3.47%)	5,238,798 (3.61%)

**Table 4 genes-13-01741-t004:** Frequencies of dinucleotide to hexanucleotide simple sequence repeats identified in three *Bursera* species.

Species	Total	Di	Tri	Tetra	Penta	Hexa
*B. cuneata*	107,270	39,769 (37.1%)	20,606 (19.2%)	31,871 (29.7%)	10,011 (9.3%)	5013 (4.7%)
*B. bipinnata*	76,766	27,915 (36.4%)	15,049 (19.6%)	23,067 (30%)	6988 (9.1%)	3747 (4.9%)
*B. palmeri*	100,614	37,480 (37.3%)	19,431 (19.3%)	29,818 (29.6%)	9212 (9.2%)	4673 (4.6%)

## Data Availability

The nuclear genomes of three *Bursera* species have been deposited in the NCBI under the project number PRJNA828589, available at https://www.ncbi.nlm.nih.gov/bioproject/PRJNA828589 (accessed on 22 September 2022).

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
