# Peer review of "Development of Genomic Resources in Mexican Bursera (Section: Bullockia: Burseraceae): Genome Assembly, Annotation, and Marker Discovery for Three Copal Species"

_genes, 2022, doi:10.3390/genes13101741_

Round 1
Reviewer 1 Report
The authors described the results of simple sequence repeats (SSRs) markers and single nucleotide polymorphisms (SNPs), based on de novo genome assembly and annotation of three Bursera species of the Bullockia section: B. bippinata, B. cuneata, and B. palmeri.
Major:
I believe the authors' idea of using NGS for de novo genome assembly and annotation of three Bursera species, for the sole purpose of identifying virtual SSR and SNP markers, is a false and expensive route. To develop SSR and SNP markers, it is necessary to use NGS analysis of multiple samples from the same species. Comparative analysis of genomes from different species is wrong, and a false path, when developing SSR and SNP markers.
Second, why go back in time, with SSR and SNP markers, when authors have NGS capability, and the need to investigate SSR and SNP markers, is no longer needed?
Third, where is the analysis of specific SSRs markers for samples from one species, for the three Bursera species of the Bullockia section: B. bippinata, B. cuneata, and B. palmeri?
For the authors, I report that the mononucleotide polymorphism does not apply to SSR markers.
2.2 Sampling and sequencing
82-83: DNA integrity was evaluated by observing a clear and unique DNA band through 2% agarose gel;
- DNA integrity is not analyzed in 2% agarose gel, DNA integrity can be checked in 1% agarose gel if necessary.
"a clear and unique DNA band" - what unique DNA means is not clear to me.
91 To obtain a set of polymorphic biallelic SNPs, we used 25 ng of high molecular weight - what does "To obtain a set of polymorphic biallelic SNPs," mean?
94 Barcoded adapters (96 forward and 96 reverse) were incorporated into the genomic - analysis of 43 Bursera individuals, the authors used 96 forward and 96 reverse barcodes, this is not correct, I guess.
Reviewer 2 Report
In the MS, the authors investigate three species of Mexican Bursera by performing draft genome assembly, annotaion, and markers-mining.
Overall, the paper is well conceived and structured.
Please, find my comments in the attached file. Specifically, I suggest the authors to deepen the markers identification paragraph, providing more informations on the identified SSRs, precenting the results relatively to the available literature.

Round 2
Reviewer 1 Report
I see that the authors have revised the article according to my recommendations. Therefore, I can recommend the paper for publication.